# Kinetics and Thermodynamics of Adsorption for Aromatic Hydrocarbon Model Systems via a Coagulation Process with a Ferric Sulfate–Lime Softening System

**DOI:** 10.3390/ma16020655

**Published:** 2023-01-10

**Authors:** Deysi J. Venegas-García, Lee D. Wilson

**Affiliations:** Department of Chemistry, University of Saskatchewan, 110 Science Place, Thorvaldson Building (Room 165), Saskatoon, SK S7N 5C9, Canada

**Keywords:** *p*-nitrophenol, naphthalene, coagulation, kinetics, thermodynamics, ferric sulfate, lime-softening, aluminum sulfate, river water

## Abstract

The adsorption mechanisms for model hydrocarbons, 4-nitrophenol (PNP), and naphthalene were studied in a coagulation-based process using a ferric sulfate–lime softening system. Kinetic and thermodynamic adsorption parameters for this system were obtained under variable ionic strength and temperature. An in situ method was used to investigate kinetic adsorption profiles for PNP and naphthalene, where a pseudo-first order kinetic model adequately described the process. Thermodynamic parameters for the coagulation of PNP and naphthalene reveal an endothermic and spontaneous process. River water was compared against lab water samples at optimized conditions, where the results reveal that ions in the river water decrease the removal efficiency (RE; %) for PNP (RE = 28 to 20.3%) and naphthalene (RE = 89.0 to 80.2%). An aluminum sulfate (alum) coagulant was compared against the ferric system. The removal of PNP with alum decreased from RE = 20.5% in lab water and to RE = 16.8% in river water. Naphthalene removal decreased from RE = 89.0% with ferric sulfate to RE = 83.2% with alum in lab water and from RE = 80.2% for the ferric system to RE = 75.1% for alum in river water. Optical microscopy and dynamic light scattering of isolated flocs corroborated the role of ions in river water, according to variable RE and floc size distribution.

## 1. Introduction

Many chemical process industries produce large volumes of oily wastewater that causes serious environmental problems when wastewater is not adequately treated [1]. In addition, oil spillage recurrently occurs during oil extraction, transportation, and storage processes, which leads to unintended soil and water contamination [2]. Hazardous chemicals are released from oil spills such as polycyclic aromatic hydrocarbons (PAHs), which are known to have harmful effects on ecosystems and human health [3]. Among the PAHs, naphthalene is relatively hydrophobic with relatively high melting/boiling points, electrochemical stability, and the propensity to reside in water, and in soils for long periods [4]. Among the various substituted arenes, *p*-nitrophenol (PNP) is regarded as a priority pollutant in industrial and agricultural wastewaters with high chemical stability, low water solubility, and high toxicity [5]. Methods such as filtration, adsorption, and coagulation are relevant for the treatment of effluent, apart from other water treatment methods [6]. Many researchers have reported the effectiveness of coagulation for the treatment of effluent with high levels of colloidal particles, organic matter, and suspended solids [7]. Coagulation is a suitable removal process for colloidal particles (negatively charged) that undergo neutralization by introduction of counter ions (positively charged) in the form of coagulants, which result in particle aggregation [8]. Chemical coagulants such as aluminum sulfate (alum), poly-aluminum chloride (PAC), alum potash, ferric sulfate or ferric chloride, among other coagulants, have been employed [9]. Four mechanisms that are either adsorption or non-adsorption processes may serve to drive coagulation [10]. Among them, inter-particle bridging and charge neutralization are the most significant factors that favour adsorption, while double layer compression and sweep flocculation are the non-adsorption factors [11]. In conventional industrial water treatment at the municipal level, lime-softening is a traditional and low-cost method used to reduce the hardness of water [12]. Lime addition to hard water (i.e., lime-softening) serves to increase the pH of water. In turn, bicarbonates transform into carbonates, where Ca^2+^ and Mg^2+^ are eliminated from water in the form of CaCO_3_ and Mg(OH)_2_ species, where such water softening is conceived for reducing the levels of calcium and magnesium ions [13]. Coagulation along with lime-softening has been proven to be an effective combination during water treatment processes for the removal of colloidal particles [14].

To reduce environmental damage to a minimum when accidental oil spills occur in aquatic environments, a scenario-specific response strategy was developed in a previous study [15]. Herein, the main aim of this work was to study the adsorption properties (kinetic and thermodynamic) of PNP and naphthalene that employs a ferric sulfate–lime softening system via a chemical coagulation process. The comparison of different coagulant systems, ferric vs. aluminum sulfate, and source water samples (laboratory vs. saline river water) was also studied for the removal of model contaminants (PNP and naphthalene). Finally, microscopy images and floc size were used to gain insight on the differences between the use of ferric sulfate over aluminum sulfate, and the effect of surface water on the properties of the flocs. 

## 2. Materials and Methods

### 2.1. Materials

All of the chemicals were of analytical reagent (AR) grade. 4-nitrophenol (99%) was obtained from Alfa Aesar (Tewksbury, MA, USA). Naphthalene (99.7%) was acquired from Mallinckrodt Pharmaceuticals (Blanchardstown, Dublin, Ireland). Potassium phosphate dibasic (98%) and potassium phosphate monobasic (99%) were purchased from Fisher Scientific (New York, NY, USA). Calcium carbonate (99%) was obtained from BDH Chemicals (Mississauga, ON, Canada). All of the stock solutions were prepared using 18 MΩ cm Millipore water (Burlington, MA, USA), unless specified otherwise. Ferric sulfate (60% *w*/*v*), lime (10% *w*/*v*) solutions, and water samples were obtained from the South Saskatchewan River water samples after primary treatment at the City of Saskatoon Water Treatment Plant (Saskatoon, SK, Canada). Aluminum sulfate (48.5% *w*/*v*) was contributed by Chemtrade Logistics (Saskatoon, SK, Canada). 

### 2.2. Kinetic Studies

Kinetic profiles for the coagulation process reflect the change in the adsorption rate and time according to the environmental conditions [16]. Kinetic adsorption equations such as the pseudo-first order (PFO) and pseudo-second order (PSO) models were used to examine the pollutant adsorption profiles [17]. The non-linear forms of the PFO and PSO kinetic models are expressed by Equations (1) and (2), respectively [18]:(1)$$q_{t}=q_{e}\left(1-exp^{-k_{1}t}\right)$$
(2)$$q_{t}=\frac{q_{e}{}^{2\;}k_{2}t}{1+q_{e}k_{2}t}$$
where *q_t_* (mg g^−1^) and *q_e_* (mg g^−1^) indicate the adsorption capacity of ferric sulfate towards pollutant removal at the time (t) and equilibrium. *k*_1_ (min^−1^) and *k*_2_ (g mg^−1^ min^−1^) are the respective rate constants for the PFO and PSO models. 

Kinetic studies were performed using the one-pot method reported previously [19]. Briefly, a 600 mL beaker was filled with 500 mL of laboratory prepared water solution with a model pollutant that was mixed via magnetic stirring. A filter paper (Whatman no. 40) was folded into a cone and attached to the beaker that was immersed into the solution to a 2 cm depth (cf. Figure 1). An initial 2.5 mL aliquot was taken from the inner filter cone which is considered as the concentration at time (t) = 0 (initial concentration), where lime was then added to the solution with a 5 min mixing time. After adding the coagulant, the sampling continued at 1 min intervals for 3 min at 295 rpm, followed by 1 min intervals for 7 min additionally at 20 rpm, then for 10 min at 2 min intervals, and finally 40 min at 5 min intervals at a 20 rpm mixing speed. Then, stirring was stopped (t = 60 min) and sampling continued for a further 40 min at 10 min intervals. Sample aliquots were prepared for UV–vis analysis, PNP λ_max_ = 400 nm and naphthalene λ_max_ = 220 nm [15]. The adsorption capacity at different times was estimated by Equation (2). To measure adsorption kinetics at variable temperatures, the one-pot experiment was carried using an Endocal refrigerated circulating bath (−40 to 40 °C) with flow control (Neslab, Newington, NH, USA) at 22, 15, and 5 °C. 

### 2.3. Thermodynamic Study of Adsorption 

Thermodynamic parameters play an important role in predicting adsorption behavior and they are strongly temperature dependent [20]. Thermodynamic parameters evaluated for PNP and naphthalene adsorption include the standard Gibbs energy change (Δ*G*°), standard enthalpy change (Δ*H*°), and the standard entropy change (Δ*S*°). These parameters were calculated using Equation (3), along with Equations (4) and (5) [21].
(3)$$\Delta G{}^{\circ}=-RTlnK_{e}$$
where *K_e_* is a thermodynamic equilibrium constant (L g^−1^) for a liquid to solution partitioning process at equilibrium, as follows: *K_e_* = *q_e_*/*C_e_*. The variable, *q_e_*, refers to the amount or concentration of the adsorbate (PNP or naphthalene) adsorbed per unit mass of the adsorbent (mg g^−1^) and *C_e_* is the equilibrium concentration of PNP or naphthalene in solution (mg L^−1^). *R* is the universal gas constant (8.314 J mol^−1^ K^−1^) and *T* is the temperature in Kelvin (K). Equation (4) outlines the relationship between Δ*G*°, Δ*H*°, and Δ*S*°, according to T (K):(4)ΔG°=ΔH°−TΔS°

Substituting Equation (3) into Equation (4), and re-arranging them gives Equation (5):(5)$$lnk_{e}=-\frac{\Delta H{}^{\circ}}{RT}+\frac{\Delta S{}^{\circ}}{R}$$

The values for the standard change for Δ*H*° and Δ*S*° were determined from the slope and intercept of the plot of *lnK_e_* versus 1/T (slope = −Δ*H*°/*R* and intercept = Δ*S*°/*R*) [22]. The standard change in Gibbs energy (Δ*G*°) for the adsorption process was calculated by employing Equation (4).

### 2.4. Coagulation Process 

The coagulation process was performed using a program-controlled conventional jar test apparatus with six 2 L jars and stirrers, at room temperature (22 °C). Approximately 1 L of water containing the model pollutant was added to the jar test setup. For lab water samples, a simulation of hardness was carried out with the standard addition of CaCO_3_. An aliquot of the simulated polluted water solution was sampled to measure the initial pollutant concentration. The coagulation experiment was carried out by adapting the procedure described by Agbovi and Wilson [10]. A predetermined amount of lime solution was added to the solution that was mixed for 20 min, where the coagulant was then added, followed by rapid stirring for 3 min at 295 rpm. Thereafter, the stirring rate was reduced to 25 rpm for 20 min. The stirring was stopped to allow the sample to settle overnight, where a 10 mL sample was centrifuged at 500 rpm for 30 min. The supernatant was analyzed by measurement of the UV−vis spectrum. PNP was buffered in order to measure concentration at pH 7. Figure 2 shows a diagram that illustrates the steps for the coagulation process and analysis. 

Optimal coagulation conditions for PNP and naphthalene removal were obtained from a previous work and are listed in Table 1. Aluminum sulfate dosage was calculated with preliminary experiments considering optimal lime dosage and changing aluminum sulfate from 1 to 100 mg L^−1^. The results reveal a higher RE (%) for PNP at 68 mg L^−1^ and for naphthalene at 75 mg L^−1^.

The removal efficiency (RE, %) and the adsorption capacity (*q_e_*) (mg·g^−1^) were calculated by Equations (6) and (7), respectively [23].
(6)$$\mathrm{RE}\;\left(\%\right)=\frac{C_{o}-C_{e}}{C_{o}}*100$$
(7)$$\;q_{e}=\frac{\left(C_{o}-C_{e}\right)*V}{m}$$

Here, *C_o_* and *C_e_* are the initial and equilibrium pollutant concentrations (mg L^−1^), *V* is the volume (L), and m is the weight (g) of the coagulant metal salt system. 

### 2.5. Optical Microscopy

Small samples of isolated wet flocs were deposited onto a microscope slide for analysis using optical microscopy. To avoid alteration of the floc structure due to compression effects, no cover slips were used during microscopy. Digital images of the flocs were captured on a Renishaw InVia Reflex Raman microscope (Renishaw plc, New Mills, UK) with 5× magnification with conventional backlighting.

### 2.6. Particle Size Distribution

The particle size of flocs that contain PNP and naphthalene were determined by dynamic light scattering (DLS) using a Malvern Zetasizer Nano ZS particle size analyzer (Worcestershire, UK). For this measurement, 0.02 mL floc suspension was diluted with 0.8 mL from the supernatant after centrifuging the floc suspension. A sample volume of 0.75 mL was used to measure the particle size in a disposable folded capillary cell (DTS1070) at 25 °C. Measured in triplicate, each measurement comprised an acquisition of 10 times and default settings for the refractive index of polystyrene particles dispersed in water were employed. 

## 3. Results and Discussion

### 3.1. Kinetics

To study the effect of ferric sulfate on pollutant removal, PNP and naphthalene were employed as model hydrocarbon pollutants for this study. The adsorption process of PNP and naphthalene can occur into two stages: a fast initial adsorption stage, where ca. 90% of the equilibrium adsorption capacity was achieved during the subsequent slow adsorption phase, where the adsorption capacity gradually reached pseudo-equilibrium conditions. The kinetic behavior of ferric sulfate for the removal of PNP and naphthalene is shown in Figure 3, where the two kinetic models (PFO and PSO) were used to fit the kinetic adsorption profiles for removal of these model pollutants. 

The corresponding best-fit kinetic parameters are listed in Table 2. The high correlation with the PFO model provides insight on the dependence of model pollutant (PNP or naphthalene) removal for the coagulation process. Based on the R^2^ values, the PNP and naphthalene uptake with ferric sulfate obeys the PFO model because of its higher R^2^ values, in contrast with the lower R^2^ values obtained by the PSO model.

To evaluate the removal of PNP and naphthalene in real water samples, it is crucial to study the effect of temperature on the kinetics for the removal of the model compounds in the river water system. Figure 4 shows the kinetics for PNP and naphthalene removal at 5, 15, and 22 °C. Kinetic results for PNP show that by decreasing the temperature from 22 to 5 °C, the adsorption capacity decreased from 8 to 4 mg g^−1^. Otherwise, naphthalene adsorption does not reveal a significant dependence with decreasing temperature (22 to 5 °C). Floc aggregation tends to be weaker when temperature decreases, because of the attenuated motional dynamics. This leads to fewer particle–particle collisions where the collision energy is low, with decreased coagulation efficiency [24].

### 3.2. Thermodynamics 

The thermodynamic behavior for PNP and naphthalene removal using the ferric sulfate–lime softening coagulant system was determined by varying the temperature from 22 to 5 °C. The thermodynamic parameters such as the standard difference in Gibbs energy (Δ*G*°), enthalpy change (Δ*H*°), and entropy change (Δ*S*°) were determined from the van ’t Hoff plot, as illustrated in Figure 5.

Table 3 shows the thermodynamic parameters Δ*G*°, Δ*H*°, and Δ*S*°. The negative value of Δ*G*° indicates the spontaneity of the process and shows that the adsorption is favorable for PNP and naphthalene, which proceeds less favorably as the temperature decreases. Negative values of the Gibbs free energy (Δ*G*°) with greater magnitude indicates a greater driving force for the adsorption, where a decrease in *q_e_* with decreasing temperature indicates a lower adsorption at lower temperatures [25]. Positive values for the standard change in enthalpy (Δ*H*°) confirmed that the process was endothermic for both systems [26]. The predominance of physical adsorption was verified by the magnitude of enthalpy change. According to Ohale et al. [27], an adsorptive process is physical in nature if the values of Δ*H*° < 80 kJ mol^−1^. The values of enthalpy obtained in this study meet this criterion: 32.7 kJ mol^−1^ for PNP and 10.8 kJ mol^−1^ for naphthalene. Positive values obtained for Δ*S*° show greater disorder (randomness) during the adsorption process.

### 3.3. Coagulation Assay with South Saskatchewan River Water and Aluminum Sulfate

This section discusses the application of the coagulant systems studied, along with real water samples from the South Saskatchewan River and the alternative use of alum (aluminum sulfate) as a coagulant. Table 4 shows the removal of PNP and naphthalene at optimal conditions for river water samples and lab water samples (prepared in Millipore water). The PNP removal using ferric sulfate as a coagulant decreases from 28.0 to 20.3% when river water was used versus laboratory prepared samples. Considering the presence of different ions and organic compounds, there exists the possibility of competition between PNP and other ions present in the river water system [28]. For samples that employ alum as a coagulant, coagulation-based removal occurs in lab water at a lower level (20.5%) compared to ferric sulfate (28.0%). The effect of using river water with alum as the coagulant results in decreased removal of PNP from 20.5 to 16.8%, as compared with lab water. The size of flocs with alum as the coagulant are small when compared with flocs obtained using ferric sulfate. When naphthalene in a lab water sample undergoes coagulation with ferric sulfate, higher removal efficiency can be achieved (89.0%). This removal decreased to 80.2% when river water was used, assuming the effect of competition relates mainly to the presence of greater ionic strength of the media. Based on an analysis of river water, the amount of dissolved solids is estimated at 250 mg/L. By comparison, the total organic carbon (TOC) was 4.9 mg/L, which indicates that the major fraction of the dissolved solids is inorganic (≈98%), in agreement with the role of prominent ionic strength effects in river water versus lab water. The use of alum as the coagulant for naphthalene removal has a lower RE (83.2%) when compared with ferric sulfate (89.0%). The same effect was observed when river water was used, as evidenced by decreasing naphthalene removal from 83.2 to 75.1%. 

### 3.4. Floc Characterization

#### 3.4.1. Floc Images

To obtain further insight into the floc properties after the coagulation process, the flocs were extracted from the bottom of the jar test beaker after precipitation. Floc images (5× magnification) were captured on a Renishaw InVia Reflex Raman microscope (Renishaw plc, New Mills, UK). Some features can be appreciated from these images. PNP flocs formed with the ferric sulfate system in lab water (Figure 6a) reveal variable particle sizes (main floc at center and small flocs around). Figure 6b shows more compacted flocs (ferric sulfate–river water). The presence of the coagulant system (ferric-lime water) with different inorganic ions (CO_3_^2−^, Cl^−^, F^−^, SO_4_^2−^, etc.), along with trace organic compounds (ca. 2%; cf. Appendix A in the Supplementary Material) could create bigger and well-defined flocs [29]. PNP flocs formed under the alum system for lab (cf. Figure 6c) or river water (cf. Figure 6d) samples present similar trends. 

Figure 7 shows flocs for naphthalene under different conditions where larger and well-defined flocs for systems using ferric sulfate as the coagulant was observed. The same behavior for PNP is observed for naphthalene, when the coagulation process was applied to river water samples (cf. Figure 7b), where bigger and denser flocs are formed. A similar behavior was observed for naphthalene flocs formed when employing the alum-based coagulant system in lab (cf. Figure 7c) or river water (cf. Figure 7d). The trend for naphthalene-based flocs is consistent with the absence of an ionic charge for this pollutant, in contrast with PNP-based flocs that possess a defined ionic charge due to ionization of PNP at the alkaline conditions employed (cf. RE (%) values in Table 4).

Flocs are a complex mixture with FeOOH, PNP or naphthalene, and calcium carbonates (total alkalinity for river water = 113 mg/L, as CaCO_3_), among other inorganic species. Hydrated iron salts can form different chemical species. The dark regions can be aggregates of ferric or calcium species due to their dominant contributions as cations, and their complexes with inorganic anions, which appeared as dark yellow under the microscope. The optical microscopy images were used to mainly identify differences in floc size and density to correlate with the kinetics of the coagulation process. In general, there appears to be a correlation between the floc density and the ionic strength upon comparison of ferric flocs in lab versus river water for PNP (cf. Figure 6a,b) and naphthalene (cf. Figure 7a,b). This trend is in accordance with the role of charge neutralization effects and the greater floc density observed for river water samples. This trend concurs with the greater ionic strength of river water and the reduced kinetic barrier for floc formation for either model pollutant (PNP or naphthalene).

#### 3.4.2. Flocs Size

The results obtained related particle size (nm) and polydispersity index (PDI) in aqueous media are presented in Table 5. The PDI is a parameter used to describe the size particle range of flocs. The measured values could be any denomination between 0 and 1, where lower values indicate more homogenous dispersions than higher values. Usually, values < 0.5 are considered acceptable concerning the homogeneity of the prepared dispersions [30]. PNP flocs with lab water showed values for PDI higher than 0.5 (ferric sulfate PDI = 0.7 ± 0.1 and aluminum PDI = 0.8 ± 0.1). These flocs are considered as non-homogeneous dispersions. PNP flocs formed with river water present for PDI values of 0.5 ± 0.1 for ferric and aluminum sulfate. These results showed that size of the flocs are more homogeneous when river versus lab water was used. This trend was attributed to the presence of greater ion levels (ca. 250 mg/L) in river water (cf. Appendix A, Supplementary Material). The trends for lab and river water corroborate the images for the floc results presented above (cf. Section 3.4.1). 

In lab water, the naphthalene flocs size presents a PDI = 1.0 for the system with ferric sulfate, but a lower value (PDI = 0.6) when alum was the coagulant. Even when both PDI values are above 0.5, naphthalene flocs formed with alum are apparently more homogenous than using ferric sulfate as the coagulant. When river water is used for naphthalene removal, the ferric sulfate PDI value is the same as that for lab water. Naphthalene uptake is lower for river water (RE% = 80.2) than lab water (RE% = 89.0) with ferric sulfate as coagulant. Based on these results, we can assume that the difference in the homogeneity of the flocs homogeneity strongly relates to the role of surface charge on the flocs and the ionic strength of the media. For the case of PNP (cf. Figure 6a,b), the ferric-based flocs in lab water tend to be more heterogeneous with lower density than those in river water. Similar trends are noted for naphthalene (cf. Figure 7a,b). Observed differences between alum and the hydrolysis products of ferric sulfate are anticipated for the adsorption mechanism of naphthalene due to the strong interactions between ferric species and carbonates. PDI differences for the alum systems concur with a naphthalene adsorption mechanism and effect of greater ion levels present in river water [31]. The greater density and homogeneity of the flocs formed in river can be related to the greater ionic strength of this media over lab water and the reduced nucleation barrier of flocs formed at higher ionic strength.

## 4. Conclusions

Coagulation of two types of model hydrocarbon oil-based compounds were studied as representative pollutants: *p*-nitrophenol (PNP) and naphthalene. Herein, an optimized ferric sulfate–lime softening coagulant system was adapted from previous work to characterize the kinetic and thermodynamic parameters of the coagulation-based removal process. The kinetic results were obtained using an in situ (one-pot) method, which showed that the PFO model provided a better fit over the PSO kinetic model for the two model pollutants. Kinetic removal (RE; %) profiles at different temperatures (5, 15, and 22 °C) were used to evaluate PNP and naphthalene. Upon decreasing the temperature from 22 to 5 °C, the adsorption capacity decreased from 8 to 4 mg g^−1^ for PNP, whereas temperature had a negligible effect on the naphthalene adsorption profile. Additionally, the coagulation process was spontaneous and endothermic in nature, based on the thermodynamic parameters. The prevalence of physical adsorption was verified by the enthalpy of the process: Δ*H*° = 32.7 kJ mol^−1^ for PNP and Δ*H*° = 10.8 kJ mol^−1^ for naphthalene. Aluminum sulfate was used as a reference coagulant to compare against the ferric sulfate system, for both river water and lab water samples, respectively. The removal efficiency (RE) of PNP with ferric sulfate system is lower for river water (RE = 20.3%) versus lab water (RE = 28.0%). By comparison, parallel trends are noted for alum, with the removal for PNP (RE = 16.8%) in river water versus PNP (RE = 20.5%) in lab water. Naphthalene coagulation follows parallel trends as noted for PNP, where the removal is generally higher overall, as follows: naphthalene with ferric sulfate as coagulant yields RE (80.2%) for river water versus lab water (RE = 89.0%). For the case of alum as the coagulant, the removal decreases from 83.2% (lab water) to 75.1% (river water). The floc images by optical microscopy and particle size analysis for PNP- and naphthalene-containing flocs show a corresponding trend related to the concentration of ions in the river water samples. The presence of greater ion concentration negatively affects the RE (%) for both ferric and aluminum systems, along with the floc size distribution, according to ionic strength effects in river water samples.

## Figures and Tables

**Figure 1 materials-16-00655-f001:**
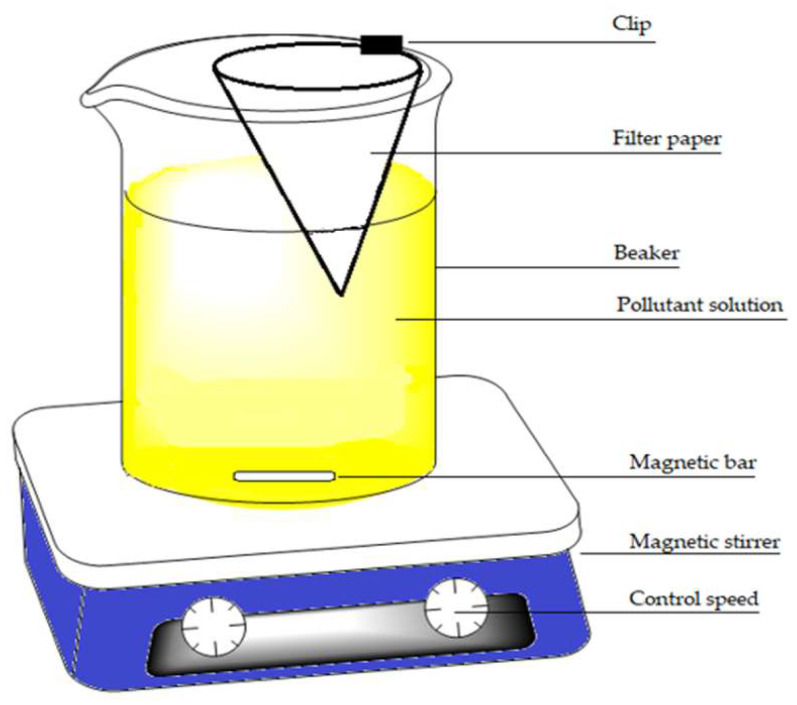
Diagram for the one-pot kinetic system.

**Figure 2 materials-16-00655-f002:**
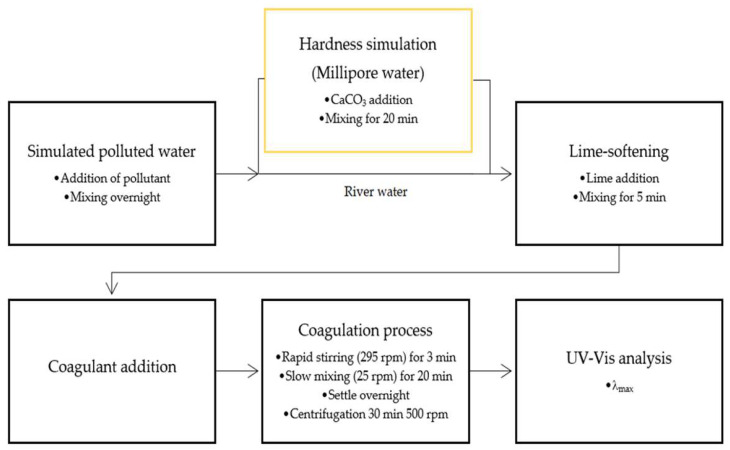
Block diagram for coagulation process for the removal of hydrocarbon oil models from lab (Millipore) versus river water samples.

**Figure 3 materials-16-00655-f003:**
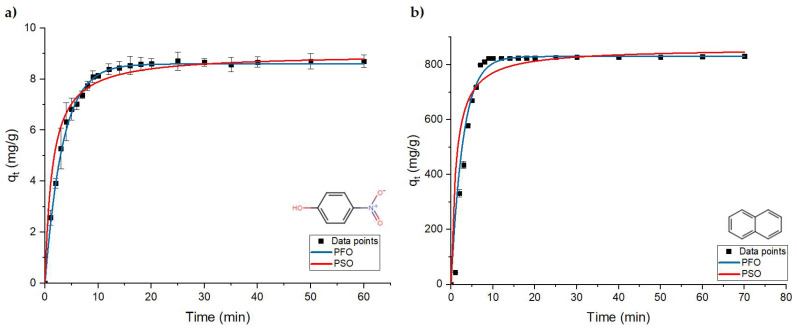
Kinetic profiles for the adsorption-based removal of model oil pollutants: (**a**) PNP and (**b**) naphthalene. The fitted lines correspond to the best-fit results for the PFO and PSO kinetic models.

**Figure 4 materials-16-00655-f004:**
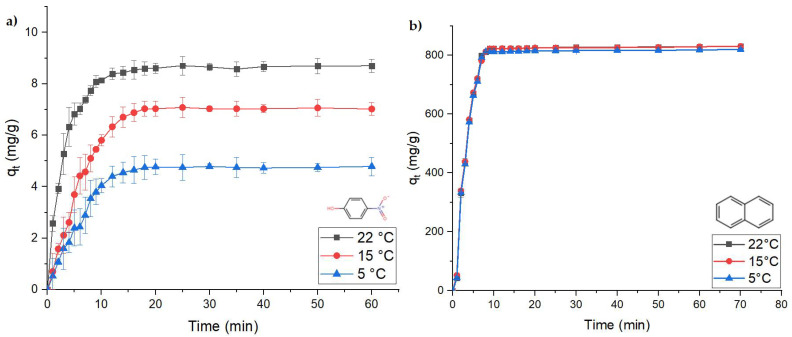
Kinetics for the adsorption of model pollutants at variable temperatures (5, 15, and 22 °C): (**a**) PNP and (**b**) naphthalene. The solid line represents a visual guide to illustrate the trend of the kinetic profile.

**Figure 5 materials-16-00655-f005:**
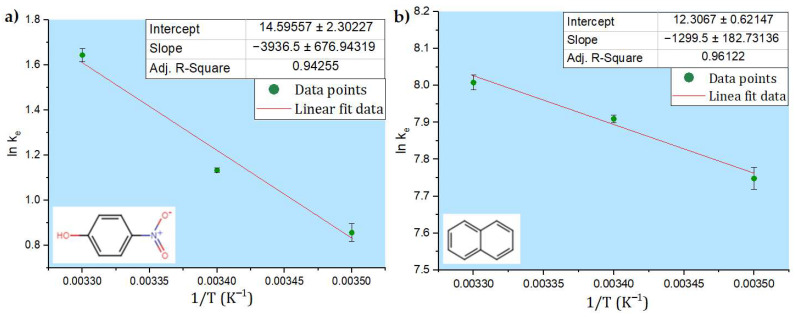
van ’t Hoff plots for the model hydrocarbons in the adsorption process: (**a**) PNP and (**b**) naphthalene.

**Figure 6 materials-16-00655-f006:**
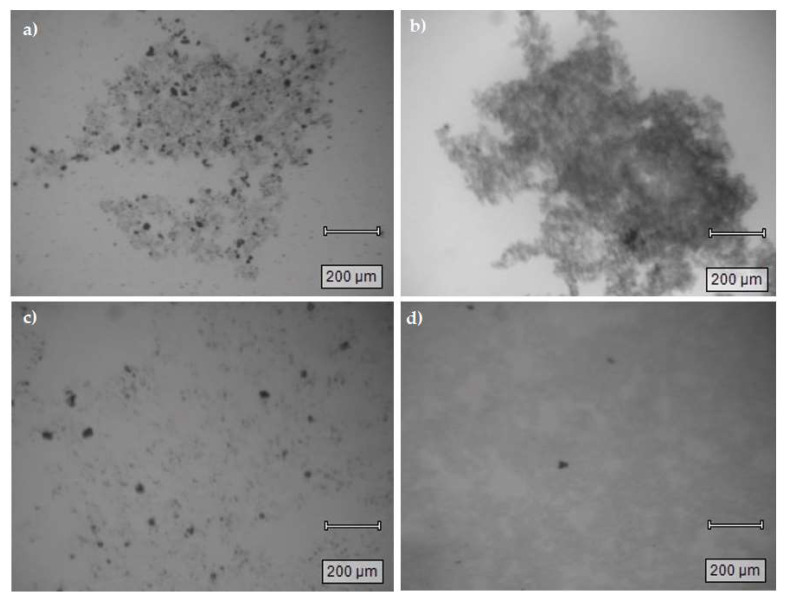
Floc sample images (5× magnification) obtained for variable coagulant systems: (**a**) PNP with lab water and ferric sulfate; (**b**) PNP with river water and ferric sulfate; (**c**) PNP with lab water and aluminum sulfate; (**d**) PNP with river water and aluminum sulfate.

**Figure 7 materials-16-00655-f007:**
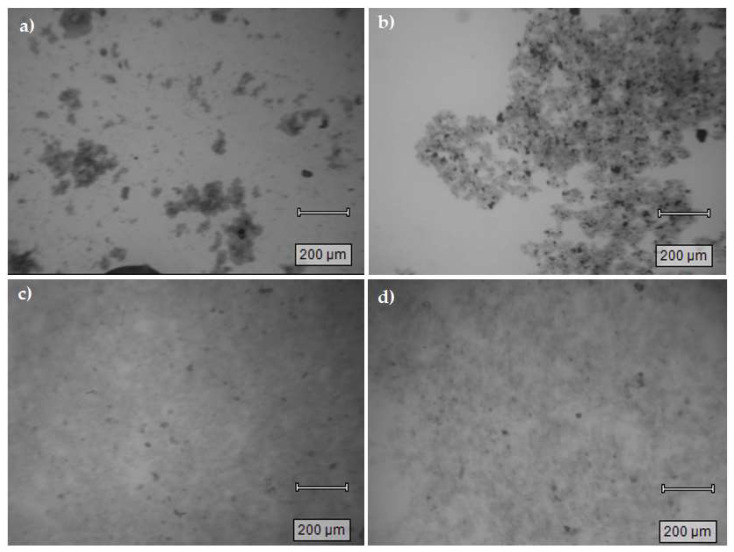
Floc sample images (5× magnification) obtained for variable coagulant systems: (**a**) naphthalene with lab water and ferric sulfate; (**b**) naphthalene with river water and ferric sulfate; (**c**) naphthalene with lab water and aluminum sulfate; (**d**) naphthalene with river water and aluminum sulfate.

**Table 1 materials-16-00655-t001:** Optimal conditions for PNP and naphthalene removal ^1^.

Model	Initial Concentration (mg L^−1^)	Ferric Sulfate (mg L^−1^)	Lime (mg L^−1^)
PNP	2	74.5	101.6
Naphthalene	16.3	42	21

^1^ Data obtained from a reported study [15].

**Table 2 materials-16-00655-t002:** Kinetic adsorption parameters for ferric sulfate coagulation process for PNP and naphthalene.

Contaminant	Pseudo-First Order Model			Pseudo-Second Order Model		
	*k* _1_	*q_e_*	R^2^	*k* _2_	*q_e_*	R^2^
	(min^−1^)	(mg g^−1^)		(min^−1^)	(mg g^−1^)	
**PNP**	0.3091 ± 0.002	8.609 ± 0.037	0.997	0.0459 ± 0.0044	9.4503 ± 0.1366	0.981
**Naphthalene**	0.3086 ± 0.031	846.9 ± 18.4	0.943	4.754 ×10^−4^ ± 1.2401 ×10^−4^	932.5 ± 39.87	0.873

**Table 3 materials-16-00655-t003:** Thermodynamic parameters for ferric sulfate coagulation process for PNP and naphthalene.

Temp (K)	(1/K)	*C_e_*	*q_e_*	*K_e_*	ln K_e_	Δ*G*°	Δ*H*°	Δ*S*°
			(mg g^−1^)	(L g^−1^)		(kJ mol^−1^)	(kJ mol^−1^)	(J mol^−1^ K^−1^)
	**PNP**							
295.15	0.0033	1.631	8.44	5.177	1.6442	−3.08	32.72	121.4
288.15	0.0034	1.700	5.28	3.106	1.1333	−2.23		
278.15	0.0035	1.868	4.40	2.355	0.8569	−1.02		
	**Naphthalene**							
295.15	0.0033	0.279	836.8	3004.6	8.0078	−19.39	10.80	102.5
288.15	0.0034	0.289	836.3	2891.7	7.9695	−18.72		
278.15	0.0035	0.360	832.9	2316.9	7.7479	−17.70		

**Table 4 materials-16-00655-t004:** Naphthalene and PNP removal efficiency (RE) with ferric and aluminum sulfate in river water samples.

Contaminant	Water	Coagulant	RE (%)
PNP	Lab	Ferric sulfate	28.0 ± 0.1
	River	Ferric sulfate	20.3 ± 0.1
	Lab	Aluminum sulfate	20.5 ± 0.3
	River	Aluminum sulfate	16.8 ± 0.4
Naphthalene	Lab	Ferric sulfate	89.0 ± 0.2
	River	Ferric sulfate	80.2 ± 0.1
	Lab	Aluminum sulfate	83.2 ± 0.3
	River	Aluminum sulfate	75.1 ± 0.5

**Table 5 materials-16-00655-t005:** Particle size distribution of floc samples that contain PNP and naphthalene model pollutants.

Sample			Size (nm)	PDI
Contaminant	Water	Coagulant		
PNP	Lab	Ferric sulfate	1950.3 ± 424.8	0.7 ± 0.1
PNP	Lab	Aluminum sulfate	2200.0 ± 717.3	0.8 ± 0.1
PNP	River	Ferric sulfate	1795.0 ± 223.3	0.5 ± 0.1
PNP	River	Aluminum sulfate	2404.3 ± 242.7	0.5 ± 0.1
Naphthalene	Lab	Ferric sulfate	2059.3 ± 1323.3	1.0 ± 0.0
Naphthalene	Lab	Aluminum sulfate	3156.7 ± 368.5	0.6 ± 0.1
Naphthalene	River	Ferric sulfate	3274.3 ± 1202.6	1.0 ± 0.0
Naphthalene	River	Aluminum sulfate	2519.7 ± 615.1	0.4 ± 0.1

## Data Availability

Not applicable.

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
