# Peer review of "Kinetics and Thermodynamics of Adsorption for Aromatic Hydrocarbon Model Systems via a Coagulation Process with a Ferric Sulfate–Lime Softening System"

_materials, 2023, doi:10.3390/ma16020655_

Round 1

Author Response

Authors’ Response to Reviewer Comments on MS ID:  materials-2070943

Please note that the Authors’ responses are in blue font. All corresponding changes were made to the manuscript in red font in the markup version of the manuscript.

Reviewer #1

This study is significant for the treatment of water pollution caused by oil spill, but there are still some parts that need to be further improved.

  1. On line 39-40, you mentioned “many researchers have reported the effectiveness of coagulation in treatment of effluent with high levels of colloidal particles, organic matter and suspended solids”, is the coagulation effectiveness of this study better than that of other studies? I suggest comparing the results of this study with some similar studies published in recent years.

Response: There is a knowledge gap in literature for PNP and naphthalene removal through coagulation. Section 1. Introduction was expanded to explain this situation.

“To reduce environmental damage to a minimum when accidental oil spills occur in aquatic environments, a scenario-specific response strategy was developed in a previous study [15], addressing a key knowledge gap related to the coagulation methodology for the removal of PNP and naphthalene.”

  1. On line70-71, I suggest adding the purities of potassium phosphate dibasic and potassium phosphate monobasic.

Response: Thanks for your constructive comment. The related information was added to provide complete description of chemicals.

  1. You choose 5℃ and 15℃ in 2.2 kinetic studies, how do you precisely control the temperature in experimental environment? And in 3.1 kinetics section, you choose 5℃, 15℃ and 22℃, could you please tell me is 22℃ the room temperature? If the answer is 'yes', how do you make sure that the temperature is same every time? If 'no', why not choose 25℃?

Response: Methodology in section 2.2 Kinetic studies was improved with the description of the instrument to control temperature.

“To measure adsorption kinetics at variable temperatures, the one pot experiment was carried using an Endocal refrigerated circulating bath (-40 °C to 40 °C) with flow control (Neslab, Newington, NH, USA) at 22 °C, 15 °C and 5 °C.”

  1. In 2.4 section, what is the temperature of coagulation process? And in line 154-155, what is the optimal lime dosage? And why do you choose this dosage?

Response: Temperature for coagulation was carried out at the ambient laboratory temperature (22 °C). Section 2.4 was improved with the following sentence:

“The coagulation process was performed using a program-controlled conventional jar test apparatus with six 2 L jars and stirrers, at the ambient room temperature (22 °C).”

The lime optimal dosage was obtained from a previous study. Information is listed in Table 1.

  1. On line 205-206. Could you please explain the reason for “kinetic results for PNP show that by decreasing the temperature from 22℃ to 5℃, the adsorption capacity decreased from 8 to 4 mg g-1” according to the adsorption mechanism? And pay attention to whether there are spaces between the number and unit.

Response:  Section 3.1 Kinetics was improved with an updated explanation for the decrease in removal efficiency.

“Floc aggregation tends to be attenuated when temperature decreases, because of the lower motion. This leads to fewer particle-particle collisions where the collision energy is low, resulting in less efficient coagulation.”

  1. What is the result of the repeated experiments of kinetic and thermodynamic studies? I think error bars could make your data more reliable.

Response: Error bars were added to kinetic and thermodynamic results.

  1. The field in real water is much larger than that in simulation system of laboratory, is it possible to completely remove PNP and naphthalene in real water?

Response: Section 3.3 Describes the coagulation of PNP and naphthalene in real water samples from Saskatchewan river, where the removal efficiency demonstrated to be lower than lab water. Greater removal can be achieved by varying the dosage of lime-water and ferric sulfate. However, the ferric sulfate and lime-water were maintained according to specified requirements based on an adapted protocol outlined in a previous study (cf. Table 2 in ref. 15) and it should be noted that the levels of contaminant employed herein are elevated relative to environmental conditions.

In summary, the authors appreciate the insightful and constructive comments provided by reviewer #1, along with the opportunity to improve the quality of this manuscript submission. We have further edited the language, syntax, and clarity throughout to meet the high standards of the journal Materials.

Reviewer 2 Report

Line 53 and 139: Use the subindex correctly for the 3 and 2 in the CaCO3 y Mg(OH)2

Line 73: What did you mean by "18 MW cm Millipore"

Why did you use a "DTS1070 cell" for measuring DLS? this cell is specially used for zeta measurements. Did you realize Z measures for the flocs or organic compounds in the water?

What temperature and the refractive index did you use for the DLS measurements?

Which wavelength and the calibration curve were used to verify the kinetics?

I highly recommend increasing the quality of the diagrams and plots used.

Suggestion: The discussion on point 3.4, probably is not a product of the ions in the water only; presumably, the different organic compounds in the river water could affect it.

I recommend doing a study identifying the presence of ions and organic compounds in the river's water, you did the retention measurements of PNP and naphthalene from the river, but you didn´t mention or show if the river´s water had any initial amount of the organic herein mentioned.

Describe the black dots in Figures 6-a and 6-c.

Author Response

Authors’ Response to Reviewer Comments on MS ID:  materials-2070943

Please note that the Authors’ responses are in blue font. All corresponding changes were made to the manuscript in red font in the markup version of the manuscript.

Reviewer # 2

Line 53 and 139: Use the subindex correctly for the 3 and 2 in the CaCO3 y Mg(OH)2

Response: The sub-indices were correctly assigned.

Line 73: What did you mean by "18 MW cm Millipore"

Response: The sentence was corrected to reflect the purity (deionization) of the water.

“All of the stock solutions were prepared using 18 MΩ · cm Millipore water, unless specified otherwise.”

Why did you use a "DTS1070 cell" for measuring DLS? this cell is specially used for zeta measurements. Did you realize Z measures for the flocs or organic compounds in the water?

Response: The disposable folded capillary cells are primarily for the measurement of zeta potential with the Zetasizer Nano series, but can be used for size measurement with ZSP, ZS and S models, according to the manufacturer specifications.

https://www.fishersci.com/shop/products/dts1070-folded-capillary-cell/NC0491866

What temperature and the refractive index did you use for the DLS measurements?

Response: Section 2.6 Particle size distribution was improved to provide complete information, where default settings were employed for the refractive index of polystyrene and a temperature of 25 °C was employed.

 “The particle size of flocs that contain PNP and naphthalene were determined by dynamic light scattering (DLS) using a Malvern Zetasizer Nano ZS particle size analyzer (Worcestershire, XZ, UK). For this measurement, 0.02 mL floc suspension was diluted with 0.8 mL from the supernatant after centrifuging the floc suspension. A sample volume of 0.75 mL was used to measure the particle size in a disposable folded capillary cell (DTS1070) at 25 °C. Measured in triplicate, each measurement comprised an acquisition of 10 times, where the default settings for the refractive index of polystyrene particles dispersed in water were used. “  

Which wavelength and the calibration curve were used to verify the kinetics?

Response: Wave numbers for PNP and naphthalene for UV-spectroscopy quantification was added in Section 2.2 Kinetic studies along with the adequate reference.

“Sample aliquots were prepared for UV-vis analysis, PNP lmax = 400 nm and naphthalene lmax = 220 nm.”

I highly recommend increasing the quality of the diagrams and plots used.

Response:  The quality of the graphs were improved to address the reviewer query.

Suggestion: The discussion on point 3.4, probably is not a product of the ions in the water only; presumably, the different organic compounds in the river water could affect it.

Response: We agree with the reviewer and added information about the organic compounds in the results section. However, based on an analysis of the river water (intake) values, the main fraction of the dissolved solids is attributed to inorganic material (ca. 98%). Based on the results in the Tables below, the dissolved solids fraction in the river water was estimated at 250 mg/L. By comparison the total organic carbon (TOC) was estimated as 4.9 mg/L. Accordingly, we infer that the major fraction of the dissolved solids are inorganic (»98%) in nature, whereas the remainder is organic (ca. 2%). Therefore, the role of ionic strength is assigned to the observed differences between lab and river water samples. 

“Considering the presence of different ions and organic compounds, there exists the possibility of competition between PNP and other ions present in the river water system.”

“This removal decreased to 80.2% when river water was used, assuming the effect of competition with ions or organic compounds that are present in river water.”

“The presence of different ions (CO3-2, Cl-, F-, SO4-2, etc.) or organic compounds could create bigger and well-defined flocs.”

I recommend doing a study identifying the presence of ions and organic compounds in the river's water, you did the retention measurements of PNP and naphthalene from the river, but you didn´t mention or show if the river´s water had any initial amount of the organic herein mentioned.

Response: To address the reviewer comment, we present a summary from the Laboratory Report for the river water obtained from the City of Saskatoon analytical lab, which is presented in Table S1 of the Supplementary Material for this study. Based on the data provided, the dissolved solids in the river water was estimated at 250 mg/L. By comparison the total organic carbon (TOC) was estimated at 4.9 mg/L. Based on this differential, we infer that the major component in the dissolved solids are of inorganic origin (»98%), which is consistent with the observations in removal efficiency for naphthalene and PNP noted above for differences in ionic strength.

The total organic carbon was 4.9 mg/L for the intake source in the South Saskatchewan River water used in this study. Additional parameters on specific classes of organics are listed in Table S1 to address the reviewer query:

Describe the black dots in Figures 6-a and 6-c.

Response: Flocs are a complex mixture with FeOOH, PNP or naphthalene, calcium carbonates (total alkalinity for river water=113 mg/L, as CaCO3), among other inorganic species. Hydrated iron salts are able to create different chemical species. These dark regions can be aggregates of ferric or calcium species due to their dominant contributions as cations, and their complexes with inorganic anions, which appeared as dark yellow under the microscope. The optical microscopy images were used to mainly identify differences in floc size and density to correlate with the kinetics of the coagulation process.  In general, there appears to be a correlation between the floc density and the ionic strength upon comparison of ferric flocs in lab versus river water for PNP (cf. Fig. 6a-b) and naphthalene (cf. Fig. 7a-b). This trend is in accordance with the role of charge neutralization effects and the greater floc density noted for river water, in agreement with its greater ionic strength and reduced kinetic barrier for floc formation for either pollutant (PNP or naphthalene).

In summary, the authors appreciate the insightful and constructive comments provided by reviewer #2, along with the opportunity to improve the quality of this manuscript submission. We have further edited the language, syntax, and clarity throughout to meet the high standards of the journal Materials.

Round 2

Reviewer 2 Report

Line 54 and 148: Please try to use the subscript instead of a number of smaller sizes.

Line 272, 303 and 332: The number reference 28, 29 and 30 shouldn't be superscript.

Figure 6 description: Please, add the description of the black dots, as I asked you in the early revision (which you describe properly in the response), and add this response into the text.

Author Response

Authors’ Response to Reviewer Comments on MS ID:  materials-2070943

Please note that the Authors’ responses are in blue font. All corresponding changes were made to the manuscript in red font in the markup version of the manuscript.

Reviewer #2

Line 54 and 148: Please try to use the subscript instead of a number of smaller sizes.

Response: Subscripts are used all through the document. If the subscript appears to be a small size letter, the appearance may be due to the font type (Palatino).

Line 272, 303 and 332: The number reference 28, 29 and 30 shouldn't be superscript.

Response: The comment addressed in the revised manuscript.

Figure 6 description: Please, add the description of the black dots, as I asked you in the early revision (which you describe properly in the response), and add this response into the text.

Response: The following response was added into the main text.

“Flocs are a complex mixture with FeOOH, PNP or naphthalene, calcium carbonates (total alkalinity for river water=113 mg/L, as CaCO3), among other inorganic species. Hydrated iron salts can form different chemical species. The dark regions can be aggregates of ferric or calcium species due to their dominant contributions as cations, and their complexes with inorganic anions, which appeared as dark yellow under the microscope. The optical microscopy images were used to mainly identify differences in floc size and density to correlate with the kinetics of the coagulation process. In general, there appears to be a correlation between the floc density and the ionic strength upon comparison of ferric flocs in lab versus river water for PNP (cf. Fig. 6a-b) and naphthalene (cf. Fig. 7a-b). This trend is in accordance with the role of charge neutralization effects and the greater floc density observed for river water samples. This trend concurs with the greater ionic strength of river water and the reduced kinetic barrier for floc formation for either model pollutant (PNP or naphthalene).”
